# Phages for Africa: The Potential Benefit and Challenges of Phage Therapy for the Livestock Sector in Sub-Saharan Africa

**DOI:** 10.3390/antibiotics10091085

**Published:** 2021-09-08

**Authors:** Angela Makumi, Amos Lucky Mhone, Josiah Odaba, Linda Guantai, Nicholas Svitek

**Affiliations:** Department of Animal and Human Health, International Livestock Research Institute (ILRI), P.O. Box 30709, Nairobi 00100, Kenya; a.mhone@cgiar.org (A.L.M.); j.odaba@cgiar.org (J.O.); l.guantai@cgiar.org (L.G.)

**Keywords:** antimicrobial resistance (AMR), multi-drug resistance (MDR), Sub-Saharan Africa (SSA), bacteriophage therapy, regulations of phage products

## Abstract

One of the world’s fastest-growing human populations is in Sub-Saharan Africa (SSA), accounting for more than 950 million people, which is approximately 13% of the global population. Livestock farming is vital to SSA as a source of food supply, employment, and income. With this population increase, meeting this demand and the choice for a greater income and dietary options come at a cost and lead to the spread of zoonotic diseases to humans. To control these diseases, farmers have opted to rely heavily on antibiotics more often to prevent disease than for treatment. The constant use of antibiotics causes a selective pressure to build resistant bacteria resulting in the emergence and spread of multi-drug resistant (MDR) organisms in the environment. This necessitates the use of alternatives such as bacteriophages in curbing zoonotic pathogens. This review covers the underlying problems of antibiotic use and resistance associated with livestock farming in SSA, bacteriophages as a suitable alternative, what attributes contribute to making bacteriophages potentially valuable for SSA and recent research on bacteriophages in Africa. Furthermore, other topics discussed include the creation of phage biobanks and the challenges facing this kind of advancement, and the regulatory aspects of phage development in SSA with a focus on Kenya.

## 1. Introduction

In Africa, a majority of the population, in a range of 250–300 million, depend on livestock for their income and livelihood, with livestock representing an average of 30% of the agricultural gross domestic product (GDP) and roughly 10% of the total GDP [1]. Animal diseases, including zoonoses, are crucial constraints in the enhancement of livestock-production systems [2] and compromise food-producing animals’ nutritional and economic potential [3]. Facing its own challenges, Africa has been reported to be one of the continents with the highest number of foodborne diseases, with approximately 91 million related diseases and 137,000 death per annum [4]. Unfortunately, on a global scale, the use of antibiotics is largely unregulated, and this is worse in developing countries where the use of antibiotics for food and animal productions to accelerate the growth of animals is rampant. Compared to other continents, Africa produces fewer antibiotics, but unregulated access and inappropriate use worsens antibiotic resistance [4,5]. Other factors, such as the poor regulation on the use of antimicrobials in both human and animals, inaccessibility to appropriate therapy, weak surveillance systems, and a lack of updated use and treatment guidelines of antimicrobials, play a role in the spread of antibiotics resistance [6]. Farmers also play a massive role in the misuse of antibiotics whereby there is a tendency to store drugs and treat animals based on symptoms they are familiar with from past infections, engaging unskilled people to treat animals, and unregulated disposal of waste in dumps. Counterfeit medicines are an additional issue that could jeopardize the fight against antimicrobial resistance [7]. Due to this constant application of antibiotics, whether for prevention, treatment, or growth promotion, this creates a selective pressure on resistant bacteria. Due to this exposure, bacteria have also developed bet-hedging strategies to resist these harsh antibiotics over time; however, this comes at survival cost for the bacteria but propels survival of a population of bacteria from extinction [7,8]. Among the strategies’ that bacteria use to acquire resistance, include the transfer of resistant genes through horizontal gene transfer, mobile genetic elements, and the bacterial toxin–antitoxin system [9].

## 2. Antibiotic Resistance in Livestock Farming

In Africa, pathogenic bacteria pose a significant challenge due to antibiotic resistance [10]. The World Health Organization (WHO) has listed priority pathogens based on research and development with a higher number of Gram-negative bacteria named as critical AMR-related threats globally [11], including *Enterobacteriaceae* that are carbapenem-resistant or extended spectrum beta-lactamase (ESBL) producing as indicated in Table 1. In other cases, resistant commensals such as *E. coli*, which rarely cause disease directly, can act as an AMR gene reservoir. The genes can be transferred to zoonotic pathogens, for example, *Salmonella enterica*, or other Gram-negative bacteria in the gut [12]. Additionally, most microorganisms are naturally transformable, for example, *C. jejuni* acquiring antibiotic-resistant genes from other organisms [13]. According to the WHO, antibiotics such as fluoroquinolones used in agricultural animals have resulted in the development of ciprofloxacin-resistant *Salmonella*, *Campylobacter*, and *E. coli*, that contribute to human infections that are difficult to treat. Tetracycline, penicillin, and sulfonamides are among the most abused antimicrobials SSA (Table 1) which has also been reported in other low resource setting countries [14]. Other bacteria species that are relevant in livestock and causes substantial loses include Methicillin-Resistant *Staphylococcus aureus*. However, it is difficult to estimate the exact prevalence of antimicrobial resistance in SSA due to the low number of antimicrobial resistance surveillance programs and research reports.

Data in Table 1 was collated from several research papers over the last ten years (2011–2021), which includes the country of research, livestock species focused on, and the sample analyzed for presence of bacteria as well as the antibiotic resistance pattern obtained from bacteria isolated. Using the data from the table above, Figure 1 below was tabulated to represent the species of livestock farmed in SSA. Additionally, poultry appears to be a popular livestock species that is commonly farmed (Figure 1) in most SSA countries but also records to be a source of high antibiotic resistance, with some African countries reporting resistance to more than ten antibiotics, as depicted in Table 1.

In comparison to other livestock farmed in SSA, but comparable to other low- and middle-income countries, the high popularity of poultry farming can be attributed to several key factors such as small body size (hence, more birds can be kept in a smaller holding), relatively short life cycle between flocks, high energy uptake efficiency, and robust adaptability to environmental conditions [50,51,52]. Due to these factors, this principally causes a shift from subsistence to intensive farming that also requires routine antimicrobial usage to improve chicken health and results in weight gain while also preventing diseases such as necrotic enteritis caused by *Clostridium perfringens* [53]. If antibiotic use is absolutely necessary, the withdrawal of antibiotics before slaughtering should be followed as the conventional standard practice; however, it could be difficult to monitor if small-scale rural poultry farmers consistently follow this guideline [54]. For this reason, alternatives need to be sought after as this will drastically decrease the overuse of antibiotics while mitigating antibiotic resistance in SSA.

## 3. Alternatives to Antibiotics Used in Livestock Farming

Given the fact that bacteria have become resistant to antibiotics and could continue to cause infections, there is a growing need to find alternatives to antibiotics in the prevention and treatment of microbial infections. One of the alternative approaches is the use of naturally occurring botanicals that can be used in place of or together with antibiotics [55]. The conventional curative system has turned its attention to traditional herbs that are rich in compounds such as alkaloids, terpenoids, tannins, steroids, and flavonoids [55]. Most traditional plants have antimicrobials that can operate in synergy with antibiotics or possess compounds that have no intrinsic antibacterial activity but can sensitize the pathogens to previously ineffective antibiotics [56].

As humans produce antibodies to fight against disease, plants also produce primary metabolites such as amino acids, fatty acids, carbohydrates, and organic acids for their survival [57]. Some of the livestock diseases that have been controlled by using alternatives include mastitis, which is the most prevalent diseases in dairy cattle worldwide [58] and with the increasing rise of antibiotic resistance in the fight against mastitis, plant extracts, essential oils, and isolated compounds are used as an alternative in treating this infection [59]. These derivatives are used to disrupt the biofilm formation, thus preventing the bacteria’s ability to adhere and multiply [60]. Other alternatives used in the control of mastitis caused by multi-drug resistant (MDR) *Staphylococcus* spp. include bacteriocin Nisin combined with dioctadecyldimethylammonium bromide (NS/DDA) nanoparticles that have been shown to be a promising treatment alternative [58]. Minthostachus verticillate is a plant that produces essential oils and limonene, a monoterpene present in the scent and resin of the plant, and these extracts have been used against *Escherichai coli*, *Bacillus pumilus*, and *Enterococcus faecium*. The essential oil from this plant is used to inhibit the growth of the pathogens, while both agents affect the formation of biofilms [61]. There are other alternatives available and currently being used, including prebiotics/probiotics, enzymes, organic acids, and plant extracts [4]; however, this is not the scope of this review.

An alternative that is going through a renaissance is the use of bacteriophages (phages), viruses that infect bacteria, which have been used and administered as pharmaceutical agents even before the discovery of antibiotics [62]. Phages are the most abundant and ubiquitous organisms on earth, and can be found in natural and man-made environments, especially those in which their bacterial host thrives [63,64]. After the discovery of antibiotics by Alexander Fleming in 1928, phage therapy was rapidly abandoned in the West. However, in countries that had witnessed the birth of phage therapy, such as Georgia and Poland, this therapy continued to flourish until modern days [65]. Phages are viruses that have the ability to infect bacteria, replicate within them, and eventually kill their susceptible host releasing progeny virions [66]. Phages use two primary life cycles to replicate, the lytic cycle and the lysogenic cycle, each having significant implications for their therapeutic application [66]. In the lytic cycle, the phage attaches itself to the bacterial cell, allowing the penetration of phage nucleic acid, transcription, translation, assembly, and exit. This exit involves killing the bacteria through the expression of endolysins and releasing multiple, as low as 20 and up to hundreds or thousands of progeny phages, which can infect other bacterial cells, thereby repeating the cycle [67]. The duration from the attachment of a phage particle to a bacterial cell and its subsequent release of new phage particles usually happens within 20–40 min but can take up to 1–2 h [68]. Due to this short life cycle, phages could potentially be used for different applications such as prior slaughter, to treat or control bacteria that may pose harm to the farmer or end user [68]. The lysogenic cycle begins with inclusion its genetic material into the chromosome of the bacterial cell, after which, replication of the phage nucleic acid together with the host genes occurs for numerous generations without major metabolic consequences for the bacterial cell, thus allowing co-existence between the phage and bacteria [69]. This also facilitates the exchange of genetic material between the phage and bacteria. However, the phage may occasionally return to the lytic cycle, leading to the release of phage particles and, in some scenarios, spreading acquired bacterial DNA [70]. Temperate phages are usually not recommended for phage therapy, as during replication they can randomly pick up a wide range of segments of bacterial DNA and transfer them to a new host. This quality makes them undesirable for therapeutic applications since virulence-associated genes, or antibiotic-resistance genes, amongst other examples, could be transferred by this route [71,72]. In some scenarios, when a suitable lytic phage cannot be isolated, it may be necessary to exclude such harmful genes, usually by synthetic biology, to circumvent or eliminate these unfavorable features. Apart from reducing undesirable qualities, other potential benefits of using synthetic biology to alter phage function include modulating the phage host range, reducing phage toxicity and immunogenicity, enhancing phage survival after administration, improving phage activity against biofilms, and enhancing bacterial killing when combined with antibiotics [71]. On the contrary to most antibiotics, phages are highly specific antibacterial agents that have the advantage of causing minor damage to the healthy microbial flora of the treated animal [73]. With the increasing cases of antimicrobial resistance worldwide, phage therapy can be used as an alternative to antibiotics and in the treatment of several bacterial infections [74]. Moreover, phages have been used to combat bacterial infections in animals with the goal of reducing the bacterial load [75].

### 3.1. Attributes of Phage-Based Products That Could Be Compelling for Livestock Farming

Livestock has been reported to be the sector contributing to resistance to the most clinically relevant antibiotics, and this paves way for the use of phages to control bacterial infections. Phages could potentially be used as additives in feed if legislation approves such products. An example of an additive is with a commercial phage product to prevent or control mastitis, *Salmonella enterica* [76], *Escherichia coli* [77], *Campylobacter* spp. [78], and *Listeria monocytogenes* [79], which are some of the most prominent foodborne zoonotic pathogens. Previous studies have shown that phages successfully reduce bacterial colonization in the gastrointestinal tract through the oral delivery route. By measuring the colony forming units (CFU), *Salmonella* phages isolated from abattoirs, chicken farms, and wastewater, administered orally to chicken in antacid suspension, successfully reduced infection by *S. Typhimurium* and *S. Enteritidis* by 4.2 log_10_ CFU and 2.19 log_10_ CFU, respectively, within 24 h [80]. *Campylobacter* phage cocktail demonstrated an ability to reduce the titer of both *C. coli* and *C. jejuni* by approximately 2 log_10_ CFU/g when administered orally and in feed for poultry [81]. The use of a broad-spectrum cocktail for phage therapy is cost-effective compared to the use of a single phage [82]. Since the discovery of phages a century ago, they have tremendously transformed modern medicine.

The efficacy of bacteriophages as antimicrobials has fostered the approval and commercialization of several products intended for the reduction of different pathogenic bacterial species [83]. Examples of some phage-related products include SalmoFree and SalmoFresh™, both containing *Salmonella enterica* phages [82,84], ListShield™ designed with *Listeria monocytogenes* phages [85], as well as phage-derived enzymes such as Lysins, integrases, and excisionases, have received considerable attention as potential antibacterial agents [86]. Phage and phage related products have advantages over antibiotics in many ways; e.g., some applications may require only a single dose since phages can self-amplify. Moreover, because phages are easy to isolate from the environment, meaning short product development time frames and reduced production costs compared to antibiotics [87], this may make them suitable for SSA. Other beneficial properties of phages include a decreased probability of resistance development if a single phage with a wide host range or a cocktail of phages is used. Additionally, phages are safe (non-toxic) for eukaryotic cells and act as a bactericidal by hijacking many essential cellular processes required by the bacteria [88]. Below, the characteristics mentioned are described in detail and may qualify them to be used in livestock.

#### 3.1.1. Single-Dose Potential

Due to their self-amplification nature and depending on the nature of the host, phages, in most cases, may not need repeated administrations over several days compared to antibiotics that require lengthy prescription doses to maintain the bioavailability of the active ingredient for a more extended period in the system [89,90]. Some studies have demonstrated that only one dose of phages could be required for therapy. A single dose of 200 µL of Avian Pathogenic *Escherichia coli* (APEC) phage cocktail (TM3, TM1, TM2, and TM4) at 10^11^ PFU/mL reduced the total viable *E. coli* count and increased the weight of chickens [91]. Polymer-encapsulated phages wV8, rV5, wV7, and wV11 reduced the shedding of *E. coli* O157:H7 in cattle feedlot upon a single dose administration [92]. In vivo studies in pigs have demonstrated that *S.* Typhimurium bacteriophages at 10^7^ PFU/mL and 10^9^ PFU/mL significantly reduced (*p* < 0.05) *S*. Typhimurium counts to 1.6 and 2.5 log_10_ CFU/mL, respectively, with a single dose after 24 h [92]. Likewise, an in vivo reduction of 1.49, 0.65, and 0.58 log_10_ CFU/mL in *Salmonella* Enteritidis number was obtained in broilers after a single oral gavage dose of 2.9 × 10^10^ PFU/mL [93]. Similarly, a single dose of vB_STy-RN5i1 and vB_STy-RN29 phages against *Salmonella enterica* obtained from market drain water was able to reduce bacterial cells by 3.1 and 2.7 log_10_ CFU/mL when characterized at 32 °C [94]. The phage cocktail (phiCcoIBB35, phiCcoIBB37, phiCcoIBB120) was able to reduce the titer of both *Campylobacter coli* and *C. jejuni* in feces by approximately 2 log_10_ CFU/g when administered by oral gavage and in feed upon a single dose [81]. The above literature indicates considerable decrease of bacterial load when single doses are used, which could be beneficial for resource-limited countries, especially if the production of large quantities of phages may not be feasible.

#### 3.1.2. Inexpensive Drugs of Infectious Diseases

Phage production is relatively cheap, while research into antibiotic discovery is highly exorbitant due to processes involved in drug discovery. Every year, about 20% of animal production losses are linked to animal infectious diseases [95]. The global animal treatment market estimates that 60–70% of farm animals in developing countries receive basic medicalization. This rate of basic medicalization is projected to increase in the coming years owing to the increase in emerging infectious diseases, antimicrobial resistance, animal welfare initiatives, and the improving regulatory framework. This increase is expected to affect the cost of production for antibiotics and other medications. The estimated cost of production of a single new antibiotic is USD 1.5 billion [96]. This is much higher than the production cost of a phage product which is between USD 8000 and USD 20,000 [97]. In the phage production model, the cost of a single dose of *Salmonella* Enteritidis is estimated at USD 0.02 [98]. With the increased funding for research and development and the low cost of production of phage-related products, it is estimated that the bacteriophage market will steadily increase in the coming years.

#### 3.1.3. Short Product Development Time Frames

Phage discovery is relatively easy because they are natural entities that are easier to isolate, purify, and characterize within a short time and at a lower cost as compared to antibiotics, which require several years of discovery and clinical trials [99]. The four methods of phages isolation—spot lysis, plaque testing, culture lysis, and routine test dilution (RTD)—have been shown to require only 24 h [100]. Likewise, the isolation of phages from animals and their environment also requires about 24 h, which is less time and effort than antibiotic discovery [100,101]. These former steps are easy to achieve but numerous factors should be taken into account in the context of product development. Bacteriophages that are considered for product development must be produced with an acceptable level of purity and have to be assessed for their efficacy in vivo and the safety of the final product. To ensure the consistency and stability of phages, the procedure for their manufacture, physicochemical and biological quality tests should be defined, as well as stringent production facilities [102]. As livestock farming in SSA is quite dynamic, with farmers rearing multiple livestock species together, this represents a complex ecology between bacteria species from different livestock as well as their interactions with phages. However, this encourages the development and delivery of local phage products that would take into account these farming dynamics.

#### 3.1.4. Decreased Probability of Resistance Development

Bacteria are less likely to develop resistance against phages when the latter are used as therapy as compared to antibiotics. One of the drawbacks to this is the host range of the phages used. The host range describes the breadth of bacteria a phage is capable of infecting [103,104,105,106]. The narrow host range which is exhibited by most phages limits the number of bacterial types with which selection for specific phage-resistance mechanisms can occur [107]. Experimental data has shown that 80% of phage-resistant variants occur mostly in wide host range phages. The use of well-characterized phage cocktails is less likely to cause phage resistance as compared to broad-spectrum antibiotics [81,108,109,110]. The reason for this phenomenon is that phage cocktails generally rely on different receptor-binding proteins during attachment, allowing specific binding of a phage to a specific host through alternate routes of entry. Using single phage preparations rather than a cocktail toward a specific bacterial species only accelerates the process of mutations, thus rendering the phage product inactive [111]. Some examples of phage cocktails that have shown high activity include the SalmoFREE^®^ phage cocktail, which demonstrated massive reduction of *Salmonella*, at the same time resulting in increased feed conversion, weight gain, homogeneity in chickens, without showing any cocktail resistant *Salmonella* strains in the course of the treatment [82]. Additionally, SalmoFresh™, which contains six strains of *Salmonella* phages, has demonstrated the capacity to reduce bacterial counts by an average of 5.34 logs CFU/mL after 5 h at 25 °C [84]. ListShield™ has demonstrated the ability to reduce the bacteria below detection levels (<10 CFU/mL) in pigs [79]. Although these phage products could be tested or used in SSA, there is need for research on local phages and their host range, which might provide more information on specificity of phages isolated in SSA, receptors involved as well as mutation frequencies of bacterial species.

## 4. Current Phage Research in Africa

Different research groups have been initiated in Africa, some of which are collaborating with Phages for Global Health (PGH). Since 2017, PGH has tasked itself with 2-week hands-on laboratory training course teaching scientists in developing countries how to isolate and characterize phages in their own regions. These countries so far include Kenya, Uganda, Ghana, and Tanzania. Using articles that have been published on different aspects of phage research, data has been collated over the last decade (2011–2021) to summarize the research that has been carried out in Africa. As this section is on the general but current phage research in Africa, the data represented herein (Table 2) include phage research on livestock, human, aquaculture, and the environment. The data in Table 2 show phage research that has been undertaken in Africa including the country, the source of phage isolation, phage host (where applicable), phage family, and the kind of research undertaken. From the table, phage research is gaining interest in Africa, whereby most research entails phage isolation and characterization, sequencing and metavirome studies. However, some gaps still exist despite many groups isolating and characterizing phages; only a few groups have delved into phage product design and development. There is also missing data on in vivo studies to test efficacy and safety of phages, phage bank establishment, and the regulatory aspect of phage products in the African context, such as that tackled in Section 6. Combining these data is important as this can help researchers from different groups across the world, and specifically the African continent, to fill in the gaps on phage research and areas that might need to be focused on. For this reason, phage research still needs to be sensitized and encouraged in Africa and collaborations with countries that have successfully used phages need to be fostered.

## 5. Hurdles of Phage Research and Regulatory Aspects of Phage Development/Products in SSA with a Focus on Kenya

As phage research in Africa is gaining interest, phages that are pure, well-characterized, sequenced, and have a defined host specificity still need to be documented. Moreover, this information should be publicly available to the different government bodies regulating veterinary practices in Africa. Currently, characterization, purification, sequencing, and storage of one phage can be achieved at a cost of about EUR 500 [162], which is not sustainable for the African continent and may need collaboration between different phage research groups around the world to cut down this cost. It is important to remember that several bacterial strains are often present in an infection; hence, multiple phage types may be needed to treat different strains of one bacterial species [162], making the goal of having a phage bank consisting of characterized phages equitably impossible if support from local governments is not achieved. Hence, our group, and several others, are pleading for the creation of phage banks across Africa to cater to the need for phages that are predicted to grow over the years to come amid the alarming rate at which AMR is progressing in the sub-continent.

Moreover, information on phage banks should become available to the local authorities, and the phage banks need to be able to supply phages for fast amplification and treatment within the shortest time possible. This creates another complication within the African context as the transport infrastructure is not well developed or may be complex within the African countries. Based on this, regional banks within Africa based on the livestock intensity or common disease outbreaks may be necessary to be able to support phage therapy in Africa. Using successful systems, for example, large phage collections already existing in Brussels, Belgium, Tbilisi, Republic of Georgia, Novosibirsk, Russia, Braunschweig, Germany, Zurich, Switzerland, Helsinki, Finland, and the Felix d’Hérelle reference phages bank in Quebec City, Canada could help in understanding the design and data storage of a phage bank [163].

Achieving a suitable phage product requires a good understanding of phage–host biology as well as phage product formulations that do not impact the effectiveness of the phage [163]. Such formulations should be well designed to fit into the African context, enabling a long shelf life, an easy to comprehend method of administration, and mechanisms in place for mass delivery in the shortest period of time [164]. So far, there is no single center in Africa serving as a manufacturing hub for bacteriophage. However, as phage research in livestock is gaining popularity in Africa, veterinary regulatory boards existing in the different countries should be alerted to support both academic and research institutes, as well as private companies, in the process of coming up with phage products. Such combined efforts between local authorities and institutions will hasten interim regulations on phage products as well as reduce stringency on phage product design.

A point to consider during the development of phage therapies for livestock that is often overlooked is the regulatory requirements and legislation aspect that can shape the end product’s design at the early stage of development. Identifying the route of administration and the relevant bacterial pathogen to target can also benefit in developing the strategy. Contrary to antibiotics legislation and regulations that have solid systems in place, phage regulation guidelines are not uniform and readily in place as a grey zone surrounds the classification of phages as biological agents, chemical agents (for enzymes derived from phages such as endolysins), veterinary medicine products, or food additive [165,166]. In the USA, phages were classified as drugs in 2011, whereas in Europe, they have been classified as medicinal products [167]. However, Georgia, one of the few countries that have maintained research and development of phage products for use in medicine, considers phages as pharmaceuticals [168]. Even in Poland, which has been a pioneer in phage therapy in Europe, phage therapy is classified as experimental treatment according to Polish law [167].

A problem that scientists and regulators alike are faced with concerning phage therapy is the lack of awareness about phage-based products or clinical data from large studies supporting phage therapy as an alternative to antibiotics. An initial discussion with the Veterinary Medicines Directorate (VMD) of Kenya indicated a great interest from this regulatory body for a phage-based product to tackle AMR in Kenya (Svitek N., personal communication). In Kenya, according to the VMD, a veterinary medicinal product includes pharmaceutical products, vaccines, alternative medicines products as well as biological products, and veterinary feed supplements [169]. Key parameters brought forward by the VMD of Kenya to assess a new product include the stability over a medium- and long-term period of time at different temperatures, efficacy, and safety of the product in the targeted animal population.

One challenge that regulators are likely to encounter is the continuous renewal of phage cocktails with novel phages to counteract the emergence of resistance in bacteria [170]. By doing this, phages need to be tested again to make sure they are lytic, do not contain toxins or AMR genes, and are safe for the animals or humans using by-products of the treated animals [171]. The regulatory framework surrounding phage licensing should be flexible enough to allow slight changes in cocktail formulations for an approved product, unless it is for a hitherto unregistered product. The current regulatory framework used for antibiotics is too long and costly to be used for phages without adapting or adjusting it [172]. Moreover, in veterinary medicine, the compassionate use of phages is not likely to be the strategy of choice, as is the case in human medicine.

In Kenya, a marketing authorization (or “registration/licensing”) is given by the VMD as approval that a veterinary medicines product can be sold and used and includes the specificities of the medicine (ex: the name of the active ingredient), the animal species for which it can be used, the indications for proper use (posology, dosage, and treatment duration) [169]. The product also requires mentioning the conditions of use (for instance, the storage conditions and the shelf life, the withdrawal period before selling the animal’s meat, and the instructions for safe use as well as the instructions for safe disposal of waste) and possible contraindications. Phage-based products would therefore require going through this process at a minimum. Other information that would be needed specifically for phages includes its impact on the environment, which is currently known to be safe, and its potential spread to neighboring farms.

Another challenge with phage products is their lack of patentability potential as is in the USA and Europe, phages cannot be patented [165,173]. However, some phage cocktails have been patented or kept as proprietary by the companies that have developed them [166]. Phage is an active treatment (because it is self-replicating) so different rules apply as the “pharmacology” of phages is different [166]. An additional concern for regulators is the co-evolving property of phages that co-evolve with their host. Furthermore, another level of complexity will be added for regulating genetically modified phages designed to evade immune recognition by the host or reduce the emergence of bacterial-resistant mutants.

## 6. Conclusions

The control of zoonotic bacteria with antibiotics marks the beginning of the arms race between the discovery of new antibiotics and bacteria. However, winning this fight and mitigating these pathogens requires an intelligent approach of staying ahead of the organisms’ ability to evolve. Using bacteriophages as an alternative contributes to a part of the solution as nature is an almost infinite phage resource. With the constant worry of bacterial evolution towards antibiotics, new phages can be isolated for most kinds of problematic bacteria, as bacteria and their phages constantly co-evolve. Additionally, the use of “cocktails” of multiple phages may reduce the probability of resistant bacteria development. As phage research is being revitalized, there needs to be a practical approach on using phages, creation of phage biobanks, and regulations on phage product development to suit the African context, creating a lasting sustainable solution.

## Figures and Tables

**Figure 1 antibiotics-10-01085-f001:**
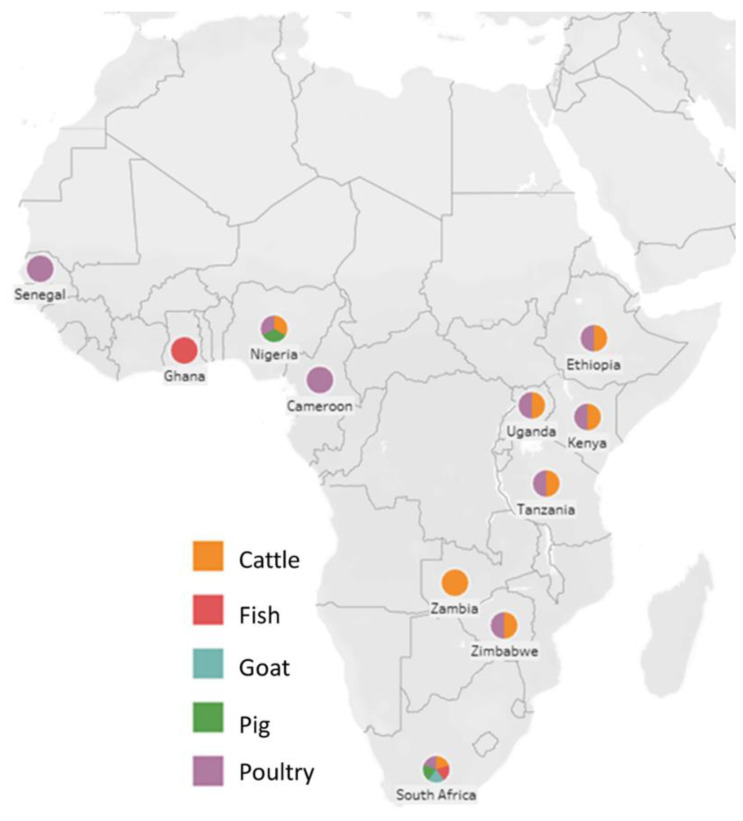
A map representing antibiotic resistance in different livestock species from the different countries, according to research studies carried out in the last decade.

**Table 1 antibiotics-10-01085-t001:** Description of antibiotic resistance in different types of bacteria isolated from livestock in SSA.

Country	Animal	Sample	Organism	Antibiotic Resistance Data	Reference
South Africa	Cattle	Milk	*S. aureus*	SPN, ERY	[15]
South Africa	Poultry	Fecal Samples	*E. coli*	CST, FLO, TRS, SPE, FOS, AMX	[16]
South Africa	Fish	Bacterial isolates	*S. aureus*	RIF(82%), CLI(82%), ERY(67%), AMP(67%), TET(27%), VAN (30%)	[17]
South Africa	Poultry	Fecal Samples	*C. jejuni*	ERY (79%), CLI (75%), AMP(54%), NAL(48%), CTR(48%), CIP(33%), GEN (15%), TET(16%)	[18]
South Africa	Poultry	Fecal Samples	*C. coli*	ERY (60%), CLI (56%),AMP (36%), NAL(26%), CTR(28%), CIP(15%), GEN (8%), TET(7%)	[18]
Kenya	Poultry	Fecal Samples	*Salmonella*	STR (6%), AMP (50%), TRS (28%), TET (11%)	[19]
Kenya	Poultry	Fecal Samples	*E. coli*	STR (9%), CHL (2%), NAL(2%), AMO (54%), TRS (26%), TET (12%)	[19]
Nigeria	Poultry	Feces, feed, water	*S. Enterica*	AMP(100%), CHL(100%), CTV(100%), CIP(100%), GEN (100%), CTA(100%), NEO(100%), NAL(100%), CPDS (100%),STR (100%), TET (100%)	[20]
Ghana	Fish	Water and cultured fish species	Coliform Bacteria	AMP(98.4%),CUR(88.9%), TET(66.7%), CTA(52.4%), TRS (56.0%), GEN (6.4%)	[21]
South Africa	Cattle	Fecal Samples	*E. coli*	ERY(63.84%),AMP(21.54%), TET(13.37%), STR(17.01%), KAN (2.42%), CHL(1.97%),NOR (1.40%)	[22]
South Africa	Cattle	Fecal Samples	Enterobacteriaceae	CAA: IMI (42%), ERT (35%), DOR (30%), MER (28%)	[23]
Uganda	Poultry	fecal samples	*Salmonella*	CIP(46.5%), SULFA(24.4%), TET(15.1%), TRI(7.0%), TRS(7.0%), CHL(4.6%), AMP(4.6%)	[24]
Ethiopia	Cattle	Milk	*E. coli*	AMP (68.7%), TRS (50%), STR (25%)	[25]
Uganda	Poultry	post-mortem samples	*E. coli*	PEN G(100%), TRS(87.5%), TET(83.9%), AMP(80.4%), AMX(69.6%), STR(67.9%), NAL(60.7%), CHL (35.7%),GEN (10.7%)	[26]
South Africa	Cattle	Fecal Samples	*Salmonella*	PEN(79%), CTA(28%), NAL(7%), CLT(24%),GEN (1%), CHL(20%), TET(62%), ERY (42%), MIN (46%), VAN (100%), OXA(100%), OFL(9%), AMP(82%), TRS(62%), STR(40%)	[27]
South Africa	Goats	Fecal samples	*Salmonella*	PEN(88%), CTA(54%), NAL(6%), CLT(37%), GEN (24%), CHL(29%), TET(32%), ERY(57%), MIN (15%), VAN(100%), OXA(100%), AMP(25%), TRS (71%), STR (29%)	[27]
Nigeria	Cattle, Poultry	Rectal and cloacol Swabs	*Enterococcus*	TET (61.0%), ERY (61.0%), QUD (4.4%), CHL (8.0%)	[28]
Senegal	Poultry	Fecal samples	*E. coli*	CST(2.2%)	[29]
South Africa	Cattle	Fecal samples	Aeromonas	AMX (100%, 92%), CHL (7%; 2%), PEN (100%; 95%), PLB (50%; 32%)	[30]
Cameroon	Poultry	muscle, liver, heart, kidney and gizzards	Various bacteria	TET (63%), KAN(45%), AMC(63%), AMP(54%), TRS (36%), ERY(81%),CTF (45%),CHL (36%), ENR (45%), GEN (54%) VAN (63%)	[31]
South Africa	Cattle	Fecal Samples	Enterococcus	VAN (100%), CLO (100%), AMI(74%), CLT (88%), STR (94%), PEN G (91%), CLI (97%), NEO (91%), ERY (99%), IMI (0.6%), AMC (8%), CIP (12%)	[32]
Zambia	Cattle	Fecal Samples	*E. coli*	CPO, CIP, AMP, TRS, TET, GEN	[33]
South Africa	Pigs/piglets, Cattle, Goats, Poultry	nasal, mouth wash, and ear swabs	*Staphylococcus*	PEN G (75%),MER (2.3%),VAN (12%),CTA (13%),CTV (40%), OXA(38%), MIN (16%),TET (83%),ERY (12%),CLI (16%),NAL (100%),CIP (3%),OFL (5%),LEV (2%)	[34]
South Africa	Fish	Water	Gram-negative bacteria	ERY (100%), AMP (85%), TRI (78%)	[35]
Nigeria	Poultry	Fecal Samples	*Salmonella*	AMP, AMC, CIP, GEN, NAL, NEO; SPE, STR, SME, TET, TRI	[36]
Ethiopia	Poultry	Eggs	*Salmonella*	CLI (100%), ERY (63%), AMP (38%), AMX (38%), TET (25%)	[37]
Tanzania	Cattle	Milk	*Staphylococcus aureus* and other bacteria	AMX, CPX, GEN, KAN, NEO, TET	[38]
Tanzania	Cattle	Fecal Samples	*E. coli*	AMP (40%), TET (20%), CTA (10%),TRS (15%)	[39]
Tanzania	Poultry	fecal Samples	*E. coli*	AMP, AMX, CHL, CIP, STR, SME, TET, TRI	[40]
Zimbabwe	Cattle	Fecal samples	*E. coli*	TET, PEN, TRS	[41]
Uganda	Cattle	Milk	*Streptococci* spp. and *Staphylococci* spp.	TET (100%)	[42]
Nigeria	Cattle and Pigs	Fecal samples	*E. coli*	PEN (96%),AMX (88%), AMP (89%), AUG (96%), CTV (58%),CTA (92%), CIX (39%), CUR (83%), CPO(58%), TET (88%), ERY (82%), STR (79%), GEN (49%), CIP(5%), OFL (5%), CLO (84%), TRS (90%), CHL (92%)	[43]
Nigeria	Poultry	Cloacae and nasal samples	*Staphylococcus aureus*	AUG(0.8%), CXI (6.1%), CUR (5.3%), CHL (12.1%),DOX (7.7%), ERY (19.4%), GEN (5.3%), LEV (0.8%), TET (45.7%),TRS (40.9%)	[44]
Zimbabwe	Poultry	Fecal samples	*E. coli*	TET (100%), BCN (100%), CLO (100%) AMP (94.1%)	[45]
Kenya	Cattle	Milk	*Staphylococcus aureus* and *Streptococcus agalactiae* and other bacteria	TRS (76%), AMP (57%)	[46]
South Africa	Cattle	Fecal samples	*E. coli*	AMP, SFZ, TET, STR	[47]
South Africa	Poultry	Isolates	*E. coli*	CST (13.5%)	[48]
Ethiopia	Cattle	Milk	*Staphylococcus* species and coliforms	AMP, ERY, NAL, CLI, TRS, CHL	[49]
South Africa	Cattle	Milk	Bacteria	PEN (47.8), OXA (1.1%), CLT (1.1%), STR (16.7%), NEO (5.6%), TET (11.1%), TRS (1.1%), ENR (1.1%), TLS (2.2%)	[49]

The antibiotic codes are as follows: CST: Colistin, TET: Tetracycline, AMX: Amoxicillin, PEN: Penicillin, AMP: Ampicillin, ERY: Erythromycin, RIF: Rifampicin, CLI: Clindamycin, AMC: Amoxicillin/Clavulanate, CIP: Ciprofloxacin, VAN: Vancomycin, AMO: Amoxicillin, NAL: Nalidixic acid, CTR: Ceftriaxone, GEN: Gentamycin, STR: Streptomycin, TRS: Trimethoprin/sulphamethoxazole, TRI: Trimethoprin, SME: Sulphamethoxazole, CHL: Chloramphenicol, CTA: Cefotaxime, KAN: Kanamycin, IMI: Imipenem, LEV: Levofloxacin, SPE: Spectinomycin, OXA: Oxacillin, AUG: Augementin, ERT: Ertapenem, MER: Meropenem, DOR: Doripenem, CAA: Carbapenem antimicrobial agents, CPDS: Compound Sulphonamindes, CUR: Cefuroxime, MIN: Minocycline, CLT: Cephalothin, ENR: Enrofloxacin, CTV: Ceftazidime, CLO: Cloxacillin, AMI: Amikacin, CIX: Cefixime, CPO: Cefpodoxime, OFL: Oflocacin, NOR: Norfloxacin, CPX: Cephalexin, NEO: Neomycin, CXI: Cefoxitin, DOX: Doxycline, FLO: Flofenicol, FOS: Fosomycin, SPN: Spiramycin, QUD: Quinupristin-dalfopristin, SFZ: Sulphufurazole, BCN: Bacitracin, PMB: PolymycinB, CTF: Ceftiofur, TLS: Tylosin, SULFA: Sulfanomide. Percentages (%) of isolates resistant to antibiotics were included where data was available. Where two different percentages are included, two different groups of isolates belonging to the same species were tested for antibiotic resistance.

**Table 2 antibiotics-10-01085-t002:** Representation of phage research in Africa over a decade (2011–2021).

Country	Source of Sample	Host	Phage	Purpose of Research	Ref.
Tanzania	Hadza fecal samples	Firmicutes	*N. I	Sequenced DNA from diverse ecosystems for phage genomes	[112]
Kenya	Baboon fecal samples	Actinobacteria, Proteobacteria, Firmicutes	*N. I	Sequenced DNA from diverse ecosystems for phage genomes	[112]
South Africa	Thiocyanate bioreactor	Proteobacteria	*N. I	Sequenced DNA from diverse ecosystems for phage genomes	[112]
South Africa	Cattle feces	Non-O157 Shiga toxin-producing *Escherichia coli* (STEC)	*Myoviridae*, *Siphoviridae*	Isolation and characterization	[113]
Kenya	Environmental water samples	*Ralstonia* *solanacearum* strain GIM1.74.	*Podoviridae*	Evolution experiments for phage stability/storage	[114]
Kenya	Lake Elmentaita sediment samples	*Vibrio metschnikovii*, *Bacillus pseudofirmus*, *Bacillus bogoriensis*, *Bacillus horikoshii*, *Bacillus cohnii*, *bacillus psedolcaliphilus*, *Bacillus halmapalus*, *Exiguobacterium* *aurantiacum, Exiguobacterium* *alkaliphilum*	*Myoviridae*, *Siphoviridae*, *Podoviridae*	Isolation, characterization, comparative genomics	[115,116]
South Africa	Skin	*Staphylococcus capitis*, *Pseudomonas*	*Myoviridae*, *Siphoviridae*, *Podoviridae*	Metaviriome analysis	[117]
Tunisia	Raw and treated wastewaters of human and animal origin	*Escherichia coli*, *Salmonella* Typhimurium, *Bact. fragilis*, *Bact. thetaiotaomicron*	Somatic coliphages *(SOMCPH)*, F-specific RNA bacteriophages *(F-RNA), Bact. fragilis* phages *(RYC2056) and Bact.* *thetaiotaomicron* phages	Monitor the microbial quality of water	[118]
South Africa	Water samples collected from taps, boreholes, and dams	*V. harveyi*, *V. parahaemolyticus*, *V. cholerae*, *V. mimicus*, *V. vulnificus*	*Myoviridae*	Isolation and characterization	[119]
South Africa	Carcass remnants	*Bacillus anthracis*	*Myoviridae*	Isolation and characterization	[120]
South Africa	Cattle feces	Shiga toxin-producing *Escherichia coli (STEC)*	*N. I	Isolation and characterization	[121]
Kenya	Lake Magadi soil sediments	*Bacillus*- and *Paracoccus* species	*Myoviridae*	Isolation and characterization	[122]
Ethiopia	Lake Chala soil sediments	*Bacillus*- and *Paracoccus* species	*Myoviridae*, *Siphoviridae*	Isolation and characterization	[122]
South Africa	Cattle feces	*Escherichia coli* O177	*Myoviridae*	Isolation and characterization	[123]
South Africa	Vaginal swabs	*Lactobacillus jensenii*, *Lactobacillus crispatus*, *Lactobacillus iners*, *Lactobacillus gasseri* and *Lactobacillus vaginalis*	*Myoviridae*, *Siphoviridae*, *Podoviridae*	Isolation and characterization	[124]
Democratic Republic of Congo	*N. I	*Salmonella* Typhi	*Myoviridae*, *Siphoviridae*, *Podoviridae*	Testing of 14 *Salmonella* phages from the Eliava collection and commercial phage cocktail “INTESTI phage”	[125]
South Africa	Cattle feces	*Escherichia coli* O157:H7	*Podoviridae*	Genome sequence	[126]
Côte d’Ivoire	Sewage water	*Achromobacter xylosoxidans*	*Siphoviridae*, *Podoviridae*	Isolation and characterization	[127]
Egypt	Chicken feces	*Salmonella* Serovars, *Citrobacter freundii*, *Enterobacter cloacae*, *Escherichia coli*.	*Siphoviridae*, *Myoviridae*	Isolation and characterization	[128]
South Africa	Human stool samples	*N. I	crAssphage	Sequencing	[129]
Kenya	Lake Victoria water samples	*Vibrio cholerae*	*Myoviridae*	Isolation and characterization	[130]
Uganda	Chicken postmortem samples	*Avian Pathogenic Escherichia coli*	*N. I	Isolation and characterization	[26]
South Africa	Cattle feces	*Escherichia coli* O177	*Myoviridae*	Efficacy of beef decontamination and biofilm disruption	[131]
Côte d’Ivoire	Sewage samples	*Pseudomonas aeruginosa*	*Myoviridae*, *Siphoviridae*, *Podoviridae*	Characterization and sequencing	[132]
South Africa	Umhlangane River water sample	*N. I	*Myoviridae*, *Siphoviridae*, *Podoviridae*	Diversity of bacteriophage population	[133]
South Africa	*N. I	*N. I	*N. I	A predator–prey model to analyze phage–bacteria interactions	[134]
Senegal	Gut and water samples of Tilapia *Sarotherodon* *melanotheron*	*N. I	*Myoviridae*, *Siphoviridae*, *Podoviridae*	Viriome analysis	[135]
South Africa	Soil samples	*N. I	*Escherichia coli bacteriophage Lambda W60*	Isolating new endonucleases using functional metagenomic techniques	[136]
South Africa	Soil samples	*N. I	*Siphoviridae*	Metaviromic techniques for viral diversity	[136]
Nigeria	Sewage water	*Pseudomonas aeruginosa*	*Myoviridae*	Genome sequencing	[137]
Nigeria	Human stool samples	*N. I	crAssphage	Quantitative CrAssphage analysis from multiple geographically distant populations	[138]
Sudan	Human stool samples	*N. I	crAssphage	Quantitative CrAssphage analysis from multiple geographically distant populations	[138]
Malawi	Water samples	*S.* Typhimurium, *S.* Enteritidis	*Ackermannviridae*, *Siphoviridae*	Isolation and characterization	[139]
Egypt	Sewage samples	*Pseudomonas aeruginosa*	*Siphoviridae*	Isolation and characterization	[140]
Egypt	Sewage samples	*Salmonella enterica*, *Escherichia coli*	*Siphoviridae*, *Myoviridae*	Applications in food safety	[141]
Egypt	Soil samples	*Ralstonia solanacearum*	*Podoviridae*	Sequencing	[142]
South Africa	Soil samples	*N. I	*Myoviridae*, *Siphoviridae*, *Podoviridae*	Metagenomic analysis of the viral community	[143]
Namibia	Wildlife carcass	*Bacillus anthracis*	*Siphoviridae*	Dissecting novel giant Siphovirus	[144]
Egypt	Soil samples	*Ralstonia solanacearum*	*Podoviridae*	Biocontrol	[145]
South Africa	Water samples from hot springs	*N. I	*Myoviridae*, *Siphoviridae*, *Podoviridae*, *Fuselloviridae*	Metavirome analysis	[146]
Egypt	Soil samples	*Streptomyces flavovirens*	*Siphoviridae*	Sequencing	[147]
Kenya	Sewage and wastewater	*Staphylococcus aureus*	N. I	Efficacy of lysis	[148]
Cameroon	Gorilla fecal samples	*N. I	*Myoviridae*, *Siphoviridae*	Microbiome analysis	[149]
South Africa	Environmental samples	*Mycobacterium smegmatis*	*Siphoviridae*	Genomics and proteomics of mycobacteriophage	[150]
South Africa	Soil samples	*N. I	*Myoviridae*, *Siphoviridae*, *Podoviridae*, *Mimiviridae*, *Phycodnaviridae*	Metaviromes of Antarctic soils	[151]
Egypt	Water samples from sewage systems	*Escherichia coli* O104: H4 *Escherichia coli* O157: H7	*Siphoviridae*, *Podoviridae*	Isolation and characterization	[152]
South Africa	Rumen fluid	*Escherichia coli* O177	*Myoviridae*	Viability of lytic phages under simulated rumen fermentation conditions	[153]
Tunisia	Sewage and waste-water treatment		Coliphages	Presence of viruses in wastewater treatment	[154]
Mauritania	Soil and water samples	*Prochlorococcus* and *Synechococcus* sp.	*Myoviridae*	Metagenomics of viruses in the desert	[155]
Tunisia	Wastewater samples	*Klebsiella pneumoniae*	*Podoviridae*	Isolation and characterization	[156]
South Africa	Water samples	*N. I	Somatic and F-RNA Phages	Phages as an indicator of fecal contamination	
Kenya	Water samples	*Arthrospira fusiformis*	cyanophages	Cyanophages affecting an African flamingo population	[157]
Kenya	Poultry feces	*Campylobacter jejuni*	*Myoviridae*	Development of spray-dried biologics	[158]
Kenya	Poultry feces	*Campylobacter jejuni* *Campylobacter coli*	*Myoviridae*	Spray-dried anti-campylobacter powder suitable for global distribution	[159]
Kenya	Poultry feces	*Campylobacter jejuni*	*Myoviridae*	Use of Trileucine and pullulan to improve anti-campylobacter bacteriophage stability	[160]
Kenya	Poultry feces	*Campylobacter jejuni*	*Myoviridae*	Lyophilization process for campylobacter bacteriophage	[161]

*N. I mean not indicated.

## Data Availability

Not applicable.

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
