# Peer review of "Phages for Africa: The Potential Benefit and Challenges of Phage Therapy for the Livestock Sector in Sub-Saharan Africa"

_antibiotics, 2021, doi:10.3390/antibiotics10091085_

Round 1
Reviewer 1 Report
Dear Authors,
The review submitted for publication touches a very important topic, related to AMR and phage therapy development. Furthermore, the interest of developing Countries in SSA for the phage treatments development represents an example of hope and encouragement to other Countries. I wish you best of luck with the development of the database and hopefully for novel bacteriophages based products to be experimented in-vivo. Overall, the review is well structured and easily readable. Please see below just a few minor issues to improve the clarity of presentation.
L 16 This sentence can be improved. “scope” does not sound right in this sentence, I would change it with something like “This reviews underlined the issue of …AMR etc etc…, proposing bacteriophages as alternatives… etc”.
L 35-37 “Antibiotics” is repeated a bit too much. Please rephrase in a more fluid sentence
L 48 Not clear, please rephrase
L49 please change “among the strategies’” with “among the strategies that…” and “include” with “there is”
L 55 The concept is clear already, please remove this sentence
L 56-57 The first half of the sentence is very unclear, please rephrase and simplify
L 57 Why do the authors talk only about gram-negatives? AMR is a problem also with gem-positives. Furthermore, in the Table 1 is mentioned several times S. aureus, which is, though, a gram- positive.
L 64-67 Please insert a reference from WHO for this statement
L 70 Sub-Saharian Africa (SSA) haÈ™ already been expressed at the beginning. The authors can refer to it as SSA for the rest of the manuscript
Figure 1 The figure is low resolution and I would enlarge the circles to make them more visible. Also, in the legend it is stated that the figure shows the antibiotic resistance, however, no antibiotics are mentioned in the figure, only livestock. Please correct either the figure or the legend for clarity. Finally, in the colours legend, put all the livestock names either with the first letter upper case or lower case
L 70-71 The authors refer to Figure 1 to explain that there are low numbers of antibiotics usage surveillance programs, however, no numbers or references to programs are mentioned
L 76-77 I can’t see from Figure 1 that “poultry is one of the livestock species that is intensely farmed”, please correct
Table 1 This table needs attention: please revise the antibiotics data column. What are the %? It is not explained in the legend (also, the legend of a table goes on top, not at the bottom of the table). Why are the % not available for all the lines? It should be explained in the legend, otherwise they should be included where missing, or all the % removed. Also, why are all the names abbreviated except for Sulfonamide? The reference 24 for that line uses SULFA as abbreviation for Sulfonamide. Eventually, check the spelling in the rest of the table “Aeromonus” “Post-Morteum” and the upper/lower cases to be consistent
L 94-97 Here a reference is needed, especially because the Authors are talking about a specific infection by C. perfringens.
L 101 “will” makes a strong statement. I would suggest to use “could” instead
L 119 remove “negatively”
L 121 and throughout the manuscript Please use italic when mentioning genus and species
L 132 “bacterial viruses” is misleading, please say “viruses that infect bacteria”
L 145 remove the second “be used”
L 152-155 The sentence here is confusing, please rephrase for clarity
L 159 remove “daughter”
L 161 “different applications” please give some examples
L 161 The lysogenic cycle of a phage starts with the inclusion of its genetic material in the chromosome of the bacterial cell, please explain this better in the sentence
L166-168 This engineering process would indeed make them lytic phages, please explain better this concept and also give a reference
L 170 Not only resistance to antibiotics, also resistance to other compounds and virulence genes. Please improve this sentence.
L 173-176 Phages are not “growth promoters”, they can just be used as additives in feed supplements, if authorised by the legislation. Please correct the sentence.
L 179 I would say more “For this reason” rather than “Moreover”
L 201-202 here the Authors are talking about phages as antimicrobials. The diagnostic kits based on phage particles are not relevant in this sentence and can be removed.
L 205-209 This part needs a sentence to link it to the following paragraphs, otherwise the reader might think that these are claims made from the authors without literature referencing.
L 209-210 This last sentence is unclear, please rephrase the concept
L 216 I would be more cautious with this statement. Sometimes studies do not take into account interfering factors that one could encounter in a farm. I would change “is required” with “could be required”
L 227 32 °C is not considered room temperature. Also “upon” makes this sentence unclear, please rephrase.
L 247-253 This is very arguable. The early testing and characterisation of phages can be easy as described in the paragraph. However, also when bacteriophages are used to develop a product, clinical trials are required to assess the efficacy in-vivo and the safety of the product. Finally, if the product works and has no side effects, the approval from the health and safety agencies is required for its use on animals (much more difficult the approval for use on humans, but this is not the topic of this review, fortunately). The tests mentioned in this paragraph are only preliminary stages. Please re-write this paragraph with more detailed information and references.
L 273 ListexP100 should be mentioned apart, because it is not a cocktail of phages, but uses only the listeriophage P100, which is a very wide spectrum one.
L 278 Please change “will” with “still”
L 283-287 Please rewrite the sentence in a clearer way
L 323 Please remove “in” before “Georgia”
L 323-325 Please give a reference at the end of this statement
L 328 Please change “alike are faced” with “are facing”
L 377-378 Please change to “reduce the probability of resistant bacteria development”
References Please check the guidelines from the Editorial board of this journal. If the title of the reference is in italic, eventual genus and species have to be non-italic and vice-versa
Author Response
L 16 This sentence can be improved. “scope” does not sound right in this sentence, I would change it with something like “This reviews underlined the issue of …AMR etc etc…, proposing bacteriophages as alternatives… etc”.
Adjusted
L 35-37 “Antibiotics” is repeated a bit too much. Please rephrase in a more fluid sentence
Rephrased
L 48 Not clear, please rephrase
Rephrased
L49 please change “among the strategies’” with “among the strategies that…” and “include” with “there is”
Changed
L 55 The concept is clear already, please remove this sentence
Removed
L 56-57 The first half of the sentence is very unclear, please rephrase and simplify
Rephrased
L 57 Why do the authors talk only about gram-negatives? AMR is a problem also with gem-positives. Furthermore, in the Table 1 is mentioned several times S. aureus, which is, though, a gram- positive- this has been adjusted
L 64-67 Please insert a reference from WHO for this statement
Reference inserted
L 70 Sub-Saharian Africa (SSA) haÈ™ already been expressed at the beginning. The authors can refer to it as SSA for the rest of the manuscript
This has been edited, Changed in whole document
Figure 1 The figure is low resolution and I would enlarge the circles to make them more visible. Also, in the legend it is stated that the figure shows the antibiotic resistance, however, no antibiotics are mentioned in the figure, only livestock. Please correct either the figure or the legend for clarity. Finally, in the colours legend, put all the livestock names either with the first letter upper case or lower case
Resolution has also been changed and the circles made bigger
L 70-71 The authors refer to Figure 1 to explain that there are low numbers of antibiotics usage surveillance programs, however, no numbers or references to programs are mentioned
Statement rephrased
L 76-77 I can’t see from Figure 1 that “poultry is one of the livestock species that is intensely farmed”, please correct
Changed statement to mean popular livestock species that is farmed alot
Table 1 This table needs attention: please revise the antibiotics data column. What are the %? It is not explained in the legend (also, the legend of a table goes on top, not at the bottom of the table). Why are the % not available for all the lines? It should be explained in the legend, otherwise they should be included where missing, or all the % removed. Also, why are all the names abbreviated except for Sulfonamide? The reference 24 for that line uses SULFA as abbreviation for Sulfonamide. Eventually, check the spelling in the rest of the table “Aeromonus” “Post-Morteum” and the upper/lower cases to be consistent
All the above have been addressed: Spelling, abbreviations and definition of the percentage
L 94-97 Here a reference is needed, especially because the Authors are talking about a specific infection by C. perfringens.
Added reference
L 101 “will” makes a strong statement. I would suggest to use “could” instead- Changed
L 119 remove “negatively”- removed
L 121 and throughout the manuscript Please use italic when mentioning genus and species- addressed
L 132 “bacterial viruses” is misleading, please say “viruses that infect bacteria”- changed
L 145 remove the second “be used”- removed
L 152-155 The sentence here is confusing, please rephrase for clarity-rephrased
L 159 remove “daughter”- removed
L 161 “different applications” please give some examples. Examples provided
L 161 The lysogenic cycle of a phage starts with the inclusion of its genetic material in the chromosome of the bacterial cell, please explain this better in the sentence- Statement added to make paragraph clear
L166-168 This engineering process would indeed make them lytic phages, please explain better this concept and also give a reference: statement added with reference
L 170 Not only resistance to antibiotics, also resistance to other compounds and virulence genes. Please improve this sentence- statement improved
L 173-176 Phages are not “growth promoters”, they can just be used as additives in feed supplements, if authorised by the legislation. Please correct the sentence. Statement has been corrected
L 179 I would say more “For this reason” rather than “Moreover”- changed
L 201-202 here the Authors are talking about phages as antimicrobials. The diagnostic kits based on phage particles are not relevant in this sentence and can be removed- This line has been removed
L 205-209 This part needs a sentence to link it to the following paragraphs, otherwise the reader might think that these are claims made from the authors without literature referencing. The statement has been changed and reference added
L 209-210 This last sentence is unclear, please rephrase the concept-statement has been rephrased
L 216 I would be more cautious with this statement. Sometimes studies do not take into account interfering factors that one could encounter in a farm. I would change “is required” with “could be required”- The paragraph has been re-written
L 227 32 °C is not considered room temperature. Also “upon” makes this sentence unclear, please rephrase- this had been rephrased
L 247-253 This is very arguable. The early testing and characterisation of phages can be easy as described in the paragraph. However, also when bacteriophages are used to develop a product, clinical trials are required to assess the efficacy in-vivo and the safety of the product. Finally, if the product works and has no side effects, the approval from the health and safety agencies is required for its use on animals (much more difficult the approval for use on humans, but this is not the topic of this review, fortunately). The tests mentioned in this paragraph are only preliminary stages. Please re-write this paragraph with more detailed information and references.- Paragraph has been re written
L 273 ListexP100 should be mentioned apart, because it is not a cocktail of phages, but uses only the listeriophage P100, which is a very wide spectrum one- deleted ListexP100
L 278 Please change “will” with “still”-changed
L 283-287 Please rewrite the sentence in a clearer way- rewritten to make statement clear
L 323 Please remove “in” before “Georgia”- removed
L 323-325 Please give a reference at the end of this statement- reference given
L 328 Please change “alike are faced” with “are facing”- changed
L 377-378 Please change to “reduce the probability of resistant bacteria development”- changed
References Please check the guidelines from the Editorial board of this journal. If the title of the reference is in italic, eventual genus and species have to be non-italic and vice-versa- Changed
Reviewer 2 Report
The manuscript of Makumi et al. has the intention of describing the advances in sub-Saharan Africa in phage therapy and to enumerate the benefits of this strategy compared to antibiotics. The English is sound, the description of resistant bacteria is useful, and the intention is good, however, the advances in phage therapy in sub-Saharan Africa are scarce and finally not described in the manuscript. I recommend to review all the African studies instead of only sub-African Africa to increase the value of this review and the number of new information since the other different sections are general and can be found in other reviews. To sum up, the lack of new data makes that this manuscript could be considered an opinion manuscript instead of a review. I recommend to add more references with new information.
Minor considerations are described below:
In section 2, it is necessary a short paragraph summarizing the data of Table 1. In addition, Table 1 should not be mentioned at line 59.
Line 63. Salmonella sp. is incorrect, it must be Salmonella enterica, please, correct this in this part and in others like this.
Section 6 describes the need of having accessibility in Africa to well-known phages since there is scarce research related to phage therapy in Africa, this section should be shortened and not an independent section, just a comment.
Lines 121, 124, 125. Please, species in italics.
Section 3. Phage therapy in livestock is too general, it is just like an introduction.
Section 4 describes basic concepts about phages instead of new concepts about the need of African research about phages.
Section 5.1. This affirmation is not totally true, there are experiments indicating non efficacy in (for instance) cattle and many examples in humans. More than one dose could be necessary to reduce bacterial counts in livestock.
Author Response
The manuscript of Makumi et al. has the intention of describing the advances in sub-Saharan Africa in phage therapy and to enumerate the benefits of this strategy compared to antibiotics. The English is sound, the description of resistant bacteria is useful, and the intention is good, however, the advances in phage therapy in sub-Saharan Africa are scarce and finally not described in the manuscript. I recommend to review all the African studies instead of only sub-African Africa to increase the value of this review and the number of new information since the other different sections are general and can be found in other reviews. To sum up, the lack of new data makes that this manuscript could be considered an opinion manuscript instead of a review. I recommend to add more references with new information.
the advances in phage therapy in sub-Saharan Africa are scarce and finally not described in the manuscript- this has been changed and a table added to showcase all phage research in Africa and not only for livestock but phage research in general.
In section 2, it is necessary a short paragraph summarizing the data of Table 1. In addition, Table 1 should not be mentioned at line 59.- paragraph introduced
Line 63. Salmonella sp. is incorrect, it must be Salmonella enterica, please, correct this in this part and in others like this- changed
Section 6 describes the need of having accessibility in Africa to well-known phages since there is scarce research related to phage therapy in Africa, this section should be shortened and not an independent section, just a comment- section 6 and 7 joined together and shortened
Lines 121, 124, 125. Please, species in italics- changed
Section 3. Phage therapy in livestock is too general, it is just like an introduction- added more information
Section 4 describes basic concepts about phages instead of new concepts about the need of African research about phages- added a few statements why this research is needed in Africa
Section 5.1. This affirmation is not totally true, there are experiments indicating non efficacy in (for instance) cattle and many examples in humans. More than one dose could be necessary to reduce bacterial counts in livestock- this has been rephrased.
Round 2
Reviewer 2 Report
The manuscript of Makumi et al. contains now more relevant information about research of phage therapy in Africa and this information is scarce in the literature. However, I also have some recommendations to be reviewed again:
-Line 59: “enterica”
-Line 68: “Figure 1” in another paragraph, delete “data from on” and continue after the period with “Poultry appears…” eliminating Furthermore.
-Lines 76-78: Delete from “Data in Table 1” to “isolates”.
-Line 136: Change the title to “current phage research in Africa”
-Delete from line 137 to line 159.
-Line 160: Delete from “As” to “(section 2)”
-Line 165: Something is missing.
-Delete complete section 5 from lines 184 to 270. This is not complete and is not the goal of the manuscript. This section should be substitute by a table and mentioned in paragraph of lines 124-135.
Author Response
Comments to the authors:
The manuscript of Makumi et al. contains now more relevant information about research of phage therapy in Africa and this information is scarce in the literature. However, I also have some recommendations to be reviewed again:
-Line 59: “enterica”- Changed
-Line 68: “Figure 1” in another paragraph, delete “data from on” and continue after the period- changed to a different paragraph and deleted as suggested
with “Poultry appears…” eliminating Furthermore- Changed
-Lines 76-78: Delete from “Data in Table 1” to “isolates”- deleted
-Line 136: Change the title to “current phage research in Africa”-changed
-Delete from line 137 to line 159- added this to the section of alternatives
-Line 160: Delete from “As” to “(section 2)”- deleted
-Line 165: Something is missing- refined the paragraph
-Delete complete section 5 from lines 184 to 270. This is not complete and is not the goal of the manuscript. This section should be substitute by a table and mentioned in paragraph of lines 124-135- Changed this section and fitted under alternatives, if this is not required or does not fit in then we could delete optionally.